# Mongolia Gerbils Are Broadly Susceptible to Hepatitis E Virus

**DOI:** 10.3390/v14061125

**Published:** 2022-05-24

**Authors:** Wenjing Zhang, Yasushi Ami, Yuriko Suzaki, Yen Hai Doan, Masamichi Muramatsu, Tian-Cheng Li

**Affiliations:** 1Department of Virology II, National Institute of Infectious Diseases, Tokyo 208-0011, Japan; zwjviolin@foxmail.com (W.Z.); muramatsu@nih.go.jp (M.M.); 2Division of Experimental Animals Research, National Institute of Infectious Diseases, Tokyo 208-0011, Japan; yami@nih.go.jp (Y.A.); ysuzaki@nih.go.jp (Y.S.); 3Center for Emergency Preparedness and Response, National Institute of Infectious Diseases, Tokyo 208-0011, Japan; yendoan@nih.go.jp

**Keywords:** hepatitis E virus, HEV, Mongolia gerbil, small-animal model

## Abstract

Although cell culture systems for hepatitis E virus (HEV) have been established by using cell lines such as PLC/PRF/5 and A549, small-animal models for this virus are limited. Since Mongolia gerbils are susceptible to genotype 1, 3 and 4 HEV (HEV-1, HEV-3 and HEV4), we intraperitoneally inoculated Mongolia gerbils with HEV-5, HEV-7, HEV-8, rabbit HEV or rat HEV in addition to the above three genotypes to investigate the infectivity and to assess whether Mongolia gerbil is an appropriate animal model for HEV infection. The results indicated that (i) HEV-5 and rat HEV were effectively replicated in the Mongolia gerbils in the same manner as HEV-4: large amounts of the viral RNA were detected in the feces and livers, and high titers of the serum anti-HEV IgG antibodies were induced in all animals. The feces were shown to contain HEV that is infectious to naïve gerbils. Furthermore, HEV-4, HEV-5 and rat HEV were successfully transmitted to the gerbils by oral inoculation. (ii) Although the viral RNA and serum anti-HEV IgG antibodies were detected in all animals inoculated with HEV-1 and HEV-8, both titers were low. The viral RNA was detected in the feces collected from two of three HEV-3-inoculated, and one of three HEV-7-inoculated gerbils, but the titers were low. The serum antibody titers were also low. The viruses excreted into the feces of HEV-1-, HEV-3-, HEV-7- and HEV-8-inoculated gerbils failed to infect naïve Mongolia gerbils. (iii) No infection sign was observed in the rabbit HEV-inoculated gerbils. These results demonstrated that Mongolia gerbils are broadly susceptible to HEV, and their degree of sensitivity was dependent on the genotype. Mongolia gerbils were observed to be susceptible to not only HEVs belonging to HEV-A but also to rat HEV belonging to HEV-C1, and thus Mongolia gerbil could be useful as a small-animal model for cross-protection experiments between HEV-A and HEV-C1. Mongolia gerbils may also be useful for the evaluation of the efficacy of vaccines against HEV.

## 1. Introduction

Hepatitis E virus (HEV), a non-enveloped virus containing a positive-sense, single-strand RNA as the genome, has been classified in the family *Hepeviridae* [1]. Although HEV infection typically results in acute and self-limited hepatitis, immunocompromised and transplant patients are vulnerable to prolonged infections and tend to develop chronic hepatitis [2]. Novel HEV strains have been identified in many animal species in recent years, and the genetic diversity within HEVs has been disclosed. As experimental data has accumulated, the existence of two genera of HEV, *Orthohepevirus* and *Piscihepevirus*, has become clear [3]. The genus *Orthohepevirus* includes at least four species: *Orthohepesvirus A*, *Orthohepesvirus B*, *Orthohepesvirus C* and *Orthohepesvirus D* (HEV-A, HEV-B, HEV-C and HEV-D) [3]. Eight viral genotypes, HEV-1 to HEV-8, have been identified in the *Orthohepesvirus A* species to date, and they are present in various animals, including human, monkey, swine, wild boar, deer, mongoose, rabbits and camels [4,5,6,7,8,9,10,11]. Although hepatitis E is caused mainly by genotypes HEV-1 to HEV-4, the number of hepatitis cases caused by novel HEVs such as HEV-7, rabbit HEV and rat HEV has recently increased [12,13,14]. Rabbit HEV is classified into an HEV-3 subtype, 3ra, in HEV-A [4]. Rat HEV belongs to genotype HEV-C1 in HEV-C, and its antigenicity and serotype differ from those of HEV-1 to HEV-8 [15,16]. HEV-5 and HEV-8 have the potential for zoonotic infection since they cross-infected primates such as cynomolgus monkeys [17,18].

Cell culture systems have been established for several HEV strains and have been used to generate infectious HEV-5, HEV-7, HEV-8 and rabbit HEV by a reverse genetics system, providing new strategies for the study of HEV biology [17,18,19,20,21]. Because most of these HEVs cross-infect cynomolgus and/or rhesus monkeys, the monkeys are useful animal models for HEV infection and vaccine development [22]. However, the costs of using monkeys are high, and recent studies indicated that most domestic rhesus monkeys and imported cynomolgus monkeys have been exposed to HEV infection, suggesting that the sources of monkeys for infection experiments are limited [23,24]. A tractable small animal remains necessary for this line of research. Although the human liver chimeric mice might be used as a small-animal model of HEV, the costs of using chimeric mice are high [25]. Since rabbits are the natural reservoir of rabbit HEV and are susceptible to HEV-4 and HEV-8 infection, and rats are susceptible to rat HEV, rabbits and rats have been used as the respective animal model for HEV infection [26,27,28]. However, rabbits and rats are not susceptible to other genotypes, and a novel small-animal model is required for HEV research.

The Mongolian gerbil (*Meriones unguiculatus*) is a small rodent in the subfamily Gerbillinae that has been used in the research of viral diseases [29,30]. Mongolian gerbils were demonstrated to be susceptible to HEV-1, -3 and -4 and were proposed as a small-animal model of HEV infection [31,32,33]. In the present study, we inoculated Mongolia gerbils with HEV-1, -3, -4, -5, -7 and -8, rabbit HEV and rat HEV, and the results demonstrated that the gerbils are broadly susceptible to HEVs. HEV-4, HEV-5 and rat HEV were efficiently replicated in the gerbils, providing new evidence of this animal as a useful animal model for these HEVs.

## 2. Materials and Methods

### 2.1. HEV Strains

We used a total of eight HEV strains in the present study. Seven strains belonging to HEV-A, i.e., HEV-1 (subtype 1a, LC061267), HEV-3 (subtype 3k, AB740232), HEV-4 (subtype 4i, LC657084), HEV-5 (AB573435), HEV-7 (KJ496144) and HEV-8 (MH410176), rabbit HEV (subtype 3ra LC484431) and one strain belonging to HEV-C, i.e., rat HEV (HEV-C1, V-105 strain, JX120573). The HEV-1 strain was obtained from a stool specimen from a cynomolgus monkey that had been experimentally infected with an Indian strain [34]. We obtained the HEV-3 strain from a pig stool specimen in a piggery, and the HEV-4 strain was collected from a wild boar caught in Japan [35]. The HEV-5, HEV-7, HEV-8 and rabbit HEV strains were produced by a reverse genetics system [17,18,20,21]. All HEV-A strains were grown in a human hepatocarcinoma cell line, PLC/PRF/5 (JCRB0406), and the cell culture supernatants were used for inoculation. The rat HEV strain was first detected in the lung tissue of a wild rat captured in Vietnam [16]. To produce large amounts of strain V-105 for infection experiments, we intravenously inoculated a Wistar rat with the 10% tissue homogenate and let it incubate for 30 days [36].

The culture supernatants or 10% stool suspensions were clarified by centrifugation at 10,000× *g* for 30 min and then passed through a 0.45 µm membrane filter (Millipore, Bedford, MA, USA). The copy numbers of the viral RNA were adjusted to 10^7^ copies/mL and stored at −80 °C until use.

### 2.2. Inoculation of Mongolia Gerbils and the Sample Collection

(i) Intraperitoneal inoculation. Six-week-old female Mongolia gerbils (MON/Jms/GbsSlc, SLC, Hamamatsu, Shizuoka, Japan) were randomly separated into nine groups (*n* = 3 per group). One group was intraperitoneally inoculated with 1 mL of phosphate-buffered saline (PBS) and used as the negative control. The remaining groups were similarly inoculated with 1 mL of the HEV-1, HEV-3, HEV-4, HEV-5, HEV-7, HEV-8, rabbit HEV or rat HEV strain containing 10^7^ copies/mL of the viral RNA. (ii) Oral inoculation. Three groups of 6-week-old female Mongolia gerbils (*n* = 3 per group) were orally inoculated with the HEV-4, HEV-5 or rat HEV strain. Five milliliters (10^7^ copies/mL) of the cell culture supernatant of HEV-4 and HEV-5 or 10% stool suspension of rat HEV was mixed within 15 mL of drinking water and then orally administered to each gerbil daily for 5 consecutive days. The administration amount of the virus was determined daily, and we confirmed that the virus mixtures were completely consumed by the gerbils.

The fecal specimens were collected 2×/week, and 10% stool suspension was used for the detection of the viral RNA. At the end of each experiment, the gerbils were euthanized by exsanguination from the heart under anesthesia, and the serum was used for the detection of the RNA, anti-HEV IgG antibodies and ALT. Tissues, including heart, liver, spleen, lung, kidney, pancreas, brain, thymus, salivary gland and muscle, were collected for the detection of the viral RNA. The tissues were washed three times with PBS and homogenized by using a MagNA Lyser (Roche, Mannheim, Germany) to prepare the 10% (*w*/*v*) tissue suspension according to the manufacturer’s recommendations.

The animal experiments were reviewed and approved by the institutional ethics committee and were performed according to the Guides for Animal Experiments at the National Institute of Infectious Diseases, Tokyo, Japan, under code 121025 (27 May 2021). All of the Mongolia gerbils were negative for the serum anti-HEV IgG antibodies by an enzyme-linked immunosorbent assay (ELISA) and negative for the HEV RNA by real-time reverse transcription-quantitative polymerase chain reaction (RT-qPCR) using the fecal specimens. The animals were individually housed in a Biosafety Level-2 facility.

### 2.3. Extraction and Detection of HEV RNA

The viral RNA was extracted from 200 µL of the samples using a MagNA Pure 96 System (Roche Applied Science, Mannheim, Germany) with a MagNA Pure 96 DNA and Viral NA Small Volume Kit (Roche Applied Science, Mannheim, Germany) according to the manufacturer’s recommendations.

A one-step RT-qPCR was carried out with a 7500 FAST Real-Time PCR System (Applied Biosystems, Foster City, CA, USA) using TaqMan Fast Virus 1-step Master Mix (Applied Biosystems, Foster City, CA, USA) under a protocol of 5 min at 50 °C, 20 s incubation at 95 °C, followed by 40 cycles of 3 s at 95 °C and 30 s at 60 °C. A forward primer JVHEVF, a reverse primer JVHEVR and a probe JVHEVP were used for the detection of HEV-1, HEV-3, HEV-4, HEV-5, HEV-7, HEV-8 and rabbit HEV [37]. A forward primer 5′-CCACGGGGGTTAATACTGC-3′ (nt 36–54), a reverse primer 5′-CGGATGCGACCAAGAAACAG-3′ (nt 189–208) and a probe 5′-FAM-CGGCTACCGCCTTTGCTAATGC-TAMRA-3′ (nt 81–102) were used for the detection of rat HEV [38]. A 10-fold serial dilution of the capped HEV-3 RNA or rat HEV RNA (10^7^ to 10^1^ copies) was used as the standard for the quantitation of the viral genome copy numbers. Amplification data were collected and analyzed with Sequence Detector software ver. 1.3 (Applied Biosystems, Foster City, CA, USA).

### 2.4. Detection of Anti-HEV IgG Antibodies

Anti-HEV IgG antibodies were detected by an ELISA using virus-like particles (VLPs) as the antigen. Since the antigenicities of HEV-1 to HEV-8 are all similar to one another and differ from that of HEV-C, we used the VLPs of rat HEV for the detection of the anti-rat HEV IgG antibodies, and we used the VLPs of HEV-1 for the detection of the anti-HEV-1, -HEV-3, -HEV-4, -HEV-5, -HEV-7, -HEV-8 and -rabbit HEV antibodies [16,39]. Briefly, flat-bottomed 96-well polystyrene microplates (Immulon 2, Dynex Technologies, Chantilly, VA, USA) were coated with 100 ng/well of the VLPs, and the duplicates of the 1:200 diluted serum samples were used. Horseradish peroxidase (HRP)-conjugated rabbit anti-Mongolia gerbil IgG antibody (H + L) (1:1000) (Bioss, Boston, MA, USA) was used as the secondary antibody.

The cut-off values were determined based on the antibody titers of 27 serum samples collected from control Mongolia gerbils before HEV inoculation. When the VLPs of HEV-1 were used, the titers ranged from 0.019 to 0.078, providing the mean OD value 0.049 with a standard deviation (SD) of 0.034, and then the cut-off was calculated as 0.15 on the basis of the mean OD plus 3× the SD (0.049  +  3 × 0.034). Similarly, the titers with the VLPs of rat HEV ranged from 0.015 to 0.081, showing the mean OD value was 0.046 with the SD of 0.036, and the cut-off was calculated as 0.15. Since HRP-conjugated anti-Mongolia gerbil IgM antibody was not available, we were unable to perform an ELISA for the detection of anti-HEV IgM antibodies.

### 2.5. Liver Enzyme Level

Alanine aminotransferase (ALT) values in the gerbil sera were monitored using a Fuji Dri-Chem Slide GPT/ALT-PIII kit (Fujifilm, Saitama, Japan). The geometric mean titer of ALT in negative control animals was considered the normal ALT titer, and a twofold or greater increase was considered a sign of ALT elevation.

### 2.6. Viral Genome Sequencing

The entire genome sequence of HEV recovered from HEV-infected Mongolia gerbils was determined by a next-generation sequencing (NGS) analysis. HEV RNA-positive fecal specimens were pooled and diluted with PBS to prepare a 10% stool suspension. The suspension was centrifuged at 10,000× *g* for 60 min and then concentrated by ultracentrifugation at 100,000× *g* for 3 h in a Beckman SW32Ti rotor. The resulting pellet was suspended in PBS at 4 °C overnight, mixed with 2.1 g CsCl, and centrifuged at 100,000× *g* for 24 h at 10 °C in a Beckman SW55Ti rotor. The gradient was fractionated into 250 μL aliquots, and the HEV RNA in each fraction was detected by RT-qPCR. The fraction with the highest HEV RNA copy number was used for the NGS analyses as described previously [40].

## 3. Results

### 3.1. Mongolia Gerbils Had Different Susceptibilities to HEV

To investigate the susceptibility of Mongolia gerbils to HEVs, the cell culture supernatant or 10% stool suspension containing 1.0 × 10^7^ copies/mL of the viral RNA was intraperitoneally inoculated into the animals, and the viral RNA in the fecal specimens was measured by RT-qPCR. As shown in Figure 1a, the viral RNAs were detected in the specimens from all of the HEV-4-inoculated, HEV-5-inoculated and rat HEV-inoculated gerbils in the early stage of inoculation, on day 4 post-inoculation (p.i.). The viral RNAs reached a peak around day 14 p.i. The titers in the peaks were substantially high, ranging from 3.3 × 10^6^ copies/g to 3.2 × 10^7^ copies/g in the HEV-4-inoculated gerbils, 5.8 × 10^7^–6.9 × 10^8^ copies/g in the HEV-5-inoculated gerbils and 6.9 × 10^7^–5.6 × 10^8^ copies/g in the rat HEV-inoculated gerbils. The viral RNA titers then decreased and were undetectable in all HEV-4-inoculated gerbils, one of the three HEV-5-inoculated gerbils and two of the three rat HEV-inoculated animals on day 28 p.i. (the end of the experimental period) (Figure 1a).

When HEV-1 and HEV-8 were used, the maximum titers were <10^5^ copies/g, although the viral RNA was detected from the fecal specimens in all of the inoculated gerbils. The viral RNA was detected in two of the HEV-3-inoculated gerbils, and one of the HEV-7-inoculated gerbils, and the copy numbers were <10^5^ copies/g. In contrast, no virus RNA was detected in the three rabbit HEV-inoculated gerbils (Figure 1a).

All gerbils were euthanized on day 28 p.i., and individual serum and tissue samples were collected. Anti-HEV IgG antibodies were detected in the serum of all animals inoculated with HEV-1, HEV-4, HEV-5, HEV-8 and rat HEV, and the titers in the HEV-4-inoculated, HEV-5-inoculated and rat HEV-inoculated animals (1:51, 200–1:409,600) were higher than those in HEV-1-inoculated and HEV-8-inoculated animals (1:800–1:6400) (Figure 1b). Two of the HEV-3-inoculated gerbils and one of the HEV-7-inoculated gerbils were positive, although the titers were low (1:1600–1:6400). The antibody was undetectable in all three rabbit HEV-inoculated animals, one of the three HEV-3-inoculated animals and two of the three HEV-7-inoculated animals (Figure 1b), and all of these gerbils were also negative for the viral RNA (Figure 1a).

As depicted in Figure 1c, the viral RNA was detected in the liver of all gerbils inoculated with HEV-1, HEV-4, HEV-5, HEV-8 or rat HEV, but the viral RNA was undetectable in the liver from the gerbils inoculated with HEV-3, HEV-7 or rabbit HEV. The viral RNA was detected from the spleen in all gerbils inoculated with HEV-4, HEV-5 or rat HEV, but the titers were lower than those in the livers (Figure 1c). No viral RNA was detected in the serum, heart, lung, kidney, pancreas, brain, thymus, salivary gland or muscle on day 28 p.i. in all animals. Neither the viral RNA nor the anti-HEV IgG antibody was detected in any of the gerbils inoculated with phosphate-buffered saline (PBS) used as negative controls.

The ALT values in the three negative control animals were 53, 57 and 60 IU/L (mean 57 IU/L), and the values in all of the HEV-infected gerbils ranged from 42 to 78 IU/L, which is less than two-fold the mean value of the negative controls (114 IU/L) (Table 1), indicating that no ALT elevation was present in the HEV-infected animals on day 28 p.i.

These results indicated that, with the exception of rabbit HEV, the Mongolia gerbils were broadly susceptible to HEV infection and showed differing susceptibility depending on the HEV genotype. HEV-4, HEV-5 and rat HEV, in particular, were efficiently replicated in the gerbils, suggesting that Mongolia gerbils have potential as a small-animal model for HEV research.

### 3.2. Analyses of the Entire Genomes of HEV-4, HEV-5, and Rat HEV

Since the Mongolia gerbil is not the original reservoir of HEV, we examined whether the extensive growth of the HEVs in this gerbil species is associated with genetic mutations during the virus replication. The viral RNA-positive stool suspensions collected from three animals in each HEV genotype group were pooled and concentrated by ultracentrifugation and used for next-generation sequencing (NGS) analyses. The entire genome nucleotide sequences of HEV-4, HEV-5 and rat HEV recovered from the fecal specimens were identical to that of each respective original nucleotide sequence. In contrast, only partial nucleotide sequences were obtained from the feces of HEV-1-, HEV-3-, HEV-7- or HEV-8-inoculated animals, although the sequences were identical to those of each original HEV. These results indicated that the replications of HEV-4, HEV-5 and rat HEV in the gerbils were genetically stable.

### 3.3. Infectivity of HEV Excreted in the Fecal Specimens

For the determination of whether the HEVs discharged into the fecal specimens collected from inoculated animals are infectious, 10% stool suspensions were prepared as described in the Materials and Methods section. The copy numbers of HEV-4, HEV-5 and rat HEV RNA in the suspension were adjusted to 1.0 × 10^5^ copies/mL. However, the highest available titers were 1.2 × 10^3^ copies/mL in HEV-1, 2.4 × 10^3^ copies/mL in HEV-3, 4.5 × 10^3^ copies/mL in HEV-7 and 1.1 × 10^3^ copies/mL in HEV-8. A total of 21 gerbils were randomly separated into seven groups (*n* = 3 per group). Each group of the animals was intraperitoneally inoculated with 1.0 mL of the suspension of these HEVs, kept and euthanized on day 42 p.i.

HEV replication was observed exclusively in the HEV-4-, HEV-5- and rat HEV-inoculated animals (Figure 2a,d,g). The viral RNA was first detected on day 4 p.i. in the HEV-4-inoculated and HEV-5-inoculated animals and on day 7–11 p.i. in the rat HEV-inoculated animals. The viral RNAs quickly increased and reached peaks with titers at 2.0 × 10^7^–4.4 × 10^8^ copies/g on day 14–18 p.i. in the HEV-4-inoculated animals, 5.5 × 10^7^–7.8 × 10^7^ copies/g on day 14 p.i. in the HEV-5-inoculated animals, and 2.0 × 10^7^–3.4 × 10^7^ copies/g on day 14–18 p.i. in the rat HEV-inoculated animals. The viral RNA copy numbers then decreased and became undetectable after day 35–39 p.i.

Serum anti-HEV IgG antibodies were detected in all HEV-4-, HEV-5- and rat HEV-inoculated animals, and the antibody titers of other animals collected on day 42 p.i. were higher than 1:12,800 (Figure 2b,e,h).

The viral RNA was detected in the liver and spleen in all animals, and the RNA copy numbers in the liver were higher than those in the spleen. No viral RNA was detected in the serum samples collected on day 42 p.i. (Figure 2c,f,i). These results indicated that the HEV-4, HEV-5 and rat HEV recovered from the gerbils were infectious.

In contrast, no sign of virus replication was observed in the HEV-1-, HEV-3-, HEV-7- and HEV-8-inoculated animals, probably because the virus titers included in the fecal specimens were too low to induce infection.

### 3.4. Detection of HEV in Tissues at the Early Stage of Infection

For the examination of the distribution of the virus at the early stage of HEV infection, gerbils were intraperitoneally inoculated with 1.0 × 10^5^ copies/mL of HEV-4 or rat HEV (*n* = 3 each) recovered from the fecal specimens of the gerbils. All animals were euthanized on day 14 p.i., and the serum, intestinal contents and tissues, including heart, liver, spleen, lung, kidney, pancreas, brain, thymus, salivary gland and muscle were collected.

The viral RNAs in the serum samples were 3.9 × 10^6^–1.1 × 10^7^ copies/mL in the HEV-4-infected gerbils and 6.8 × 10^7^–1.5 × 10^8^ copies/mL in the rat HEV-infected gerbils, suggesting that viremia occurred at the early stage of infection. High titers of viral RNA were observed in the intestinal contents of both the HEV-4-infected gerbils (2.6 × 10^7^–6.3 × 10^8^ copies/g) and rat HEV-infected gerbils (4.9 × 10^8^–7.9 × 10^8^ copies/g) (Figure 3). In all animals, the viral RNA was detected in the liver, spleen, heart, lung, kidney, pancreas, salivary gland and muscle but was not detected in the thymus or brain.

The titers of viral RNA detected in the liver (1.6 × 10^9^–1.6 × 10^10^ copies/g) and spleen (3.2 × 10^7^–4.6 × 10^8^ copies/g) from the HEV-4-infected gerbils were higher than those detected in other tissues (<4.6 × 10^5^ copies/g). Similarly, the titers of viral RNA detected in the liver (1.2 × 10^10^–2.7 × 10^10^ copies/g) and spleen (2.8 × 10^8^–2.9 × 10^9^ copies/g) from the rat HEV-infected gerbils were higher than those in other tissues (<1.9 × 10^6^ copies/g). The viral RNA was not detected in the pancreas or muscle of one gerbil or in the salivary gland of another gerbil among the rat HEV-infected animals (Figure 3b). As the amount of viral RNA in the serum was extremely high at this early stage of infection, the complete removal of the blood from the tissues by heart exsanguination seemed difficult, and the possibility of contamination from the remaining serum was present.

The ALT level was 58–98 IU/L in the HEV4-infected gerbils and 61–92 IU/L in the rat HEV-infected gerbils (Table 2), and we concluded that no ALT elevation had occurred in these animals on day 14 p.i.

### 3.5. Transmission of HEV-4, HEV-5, and Rat HEV by an Oral Inoculation

Since HEV is naturally transmitted via an oral-fecal route, we examined whether HEV transmits to Mongolia gerbils through an oral route. Three groups of gerbils, each containing three animals, were orally inoculated with HEV-4, HEV-5 or rat HEV, as described in the Materials and methods. All nine gerbils were euthanized on day 26 p.i. and the serum, bile, liver and spleen were collected. In addition, all animals fasted one day before operation to collect bile samples.

When the gerbils were inoculated with HEV-4, 1.1 × 10^4^–2.2 × 10^5^ copies/g of the viral RNAs were first detected on day 9 p.i. in fecal specimens (Figure 4a). The viral RNA of one of the three gerbils was gradually increased to 1.3 × 10^6^ copies/g on day 26 p.i. (Figure 4a), and was detected in the liver (2.0 × 10^7^ copies/g), spleen (4.3 × 10^4^ copies/g), bile (1.3 × 10^8^ copies/mL) and serum (4.3 × 10^3^ copies/mL) samples on day 26 p.i. (Figure 1c). The viral RNAs of the other two animals decreased rapidly and became undetectable after day 16 p.i. (Figure 4a), and no viral RNA was detected in the liver, spleen, bile and serum samples of these two gerbils on day 26 p.i. No serum anti-HEV IgG antibodies were detected in three animals on day 26 p.i. (Figure 4b).

In HEV-5-inoculated animals, 3.8 × 10^3^–2.3 × 10^4^ copies/g of the viral RNAs were first detected on day 12 p.i. and increased to 2.4 × 10^7^–5.4 × 10^7^ copies/g on day 26 p.i. (Figure 4d). The serum IgG antibodies counting (1:12,800–1:51,200) were detected in three animals on day 26 p.i. (Figure 4e). The viral RNAs, 4.1 × 10^6^–5.4 × 10^7^ copies/g in the liver, 1.4 × 10^5^–3.3 × 10^5^ copies/g in the spleen, 9.6 × 10^4^–2.7 × 10^7^ copies/mL in the bile, and 1.1 × 10^4^–2.2 × 10^4^ copies/mL in the serum were detected (Figure 4f).

In rat HEV-inoculated gerbils, 2.1 × 10^5^ copies/g, 3.1 × 10^5^ copies/g and 1.3 × 10^6^ copies/g of the viral RNA was first detected on day 12, 19 and 23 p.i., and increased 2.8 × 10^7^–5.4 × 10^8^ copies/g on day 26 p.i. (Figure 4g). The serum IgG antibodies counting 1:3200–1:25600 were detected in three animals on day 26 p.i. (Figure 4h). The viral RNAs, 2.2 × 10^9^–6.3 × 10^10^ copies/g) in liver, 2.6 × 10^8^–5.5 × 10^9^ copies/g in spleen, 1.2 × 10^8^–1.9 × 10^10^ copies/mL in bile and 2.8 × 10^6^–3.0 × 10^7^ copies/mL in serum were detected on day 26 p.i. (Figure 4i).

The ALT levels measured on day 26 p.i. were 47 to 70 IU/L, and no ALT elevation was observed (Table 3). These results indicated that HEV-4, HEV-5 and rat HEV were transmitted to Mongolia gerbils by an oral-fecal route.

## 4. Discussion

We selected eight HEV strains, including six genotypes belonging to HEV-A and one genotype belonging to HEV-C1, to examine the strains’ infectivity in Mongolia gerbils. The results of our experiment confirmed that Mongolia gerbils are broadly susceptible to HEV infection, and that the infectivity was considerably different among the HEV genotypes. HEV-4, HEV-5 and rat HEV effectively replicated in the Mongolia gerbils, whereas the other HEVs, including HEV-1, -3, -7 and -8 and rabbit HEV, did not. High copy numbers of the viral RNA were detected in the serum, feces, spleen and liver specimens from the HEV-4-, HEV-5- and rat HEV-infected gerbils. The viruses recovered from the feces of the infected gerbils were infectious to naïve gerbils.

We recently confirmed that cynomolgus and rhesus monkeys are susceptible to rat HEV, but the virus replication in the monkeys was limited [41]. In addition, the number of sources of monkeys is limited, and the cost of using monkeys is very high. No small animals that are susceptible to both HEV-A and HEV-C1 have been reported. Our present findings indicate that Mongolia gerbils (i) would be a valuable model for investigations of the cross-protection between HEV-A and HEV-C1, and (ii) could be useful for examination of vaccines’ efficacy against HEV-A and HEV-C1.

We observed that HEV-1 and HEV-8 caused infection in gerbils, but the viruses released in the fecal specimens were lower than 10^5^ copies/g and failed to infect naïve gerbils. These results are in contrast to those obtained in HEV-4, HEV-5 and rat HEV and suggest that the replications of HEV-1 and HEV-8 are limited in Mongolia gerbils. We thus acknowledge that Mongolia gerbils are not necessarily an appropriate animal model for HEV-1 and HEV-8. Since no signs of infection were observed in the rabbit HEV-inoculated gerbils and since not all gerbils became infected with HEV-3 and HEV-7 (Figure 1), Mongolian gerbils are also not suitable as an animal model for these HEVs.

HEV-4, HEV-5 and rat HEV were orally transmitted to the naïve gerbils, providing another inoculation route for the viruses. However, the infection indicators of the individual animals, including the virus RNA-positive period (Figure 4a), incubation period (Figure 4g) and serum IgG titers (Figure 4e,h), were more conspicuous in the animals infected by oral administration. In addition, the copy numbers of the viral RNA in the fecal specimens of the orally inoculated animals were lower than those in the intraperitoneally inoculated animals (Figure 1a, Figure 2a,d,g, Figure 4a,d,g), and no serum IgG antibody was detected in the animals orally inoculated with HEV-4 (Figure 4b), suggesting the infection efficiency was lower by oral inoculation than by the intraperitoneal administration.

The results of our NGS analyses demonstrated that the entire genomes of HEV-4, HEV-5 and rat HEV recovered from the infected gerbils had no nucleotide sequence changes, suggesting that the replication of these HEVs in the gerbils was genetically stable. Although the cell culture of HEVs has been established and is used as a convenient isolation method, genetic mutations have been apt to occur, and the effects on the virus replication and the infectivity of the recovered viruses are difficult to predict [17,20,42,43,44]. Mongolia gerbils might be valuable for avoiding unexpected mutations and for maintaining the original genome sequence during the isolation of HEVs.

Extrahepatic manifestations associated with HEV, including renal diseases and reproductive system disorders as well as pancreatitis plus a variety of neurological disorders, have been documented [45,46,47]. However, direct evidence of the linkage between these manifestations and HEV infection is lacking. It is therefore important to determine whether HEV RNA appears in tissues other than the liver of the infected animals. In the present study, low copy numbers of the HEV RNA were detected in heart, lung, kidney, pancreas, salivary gland and muscle of the HEV-4-infected and rat HEV-infected gerbils during the viremia period (Figure 3), but we could not rule out the possibility of contamination from the remaining serum in the RNA-positive tissues, and we were unable to create the corresponding figures. In contrast, high copy numbers of HEV RNA were detected in the liver and spleen not only during the viremia period (Figure 3 and Figure 4c,f,i) but also after the viremia period and virus shedding (Figure 1c and Figure 2c,f,i), demonstrating that the RNA-positive periods in the liver and spleen were longer than those in the sera and fecal specimens. It remains unclear whether the HEV was replicating in the spleen.

Our present findings indicate that the Mongolia gerbil could be a candidate small-animal model for HEV-A and HEV-C1. Unlike rats, gerbils have a gallbladder, and this is a great advantage in observing whether the HEVs are released into the bile. The use of 1-day fasting might be useful to efficiently collect the bile samples since the gerbil gallbladder is very small. We collected approx. 10 µL bile samples from the gerbils orally inoculated with HEV-4, HEV-5 and rat HEV, and we observed that the titer of the viral RNA in the bile was high (Figure 4c,f,i). The collection of the blood specimens from the gerbils, especially plural collections, was considerably difficult. We found that the measurement of the temporal changes of the viral RNA, serum antibodies and ALT in the individual gerbils was a challenging task. We collected the serum samples exclusively at the end of each experiment, and we were unable to perform kinetic experiments to measure the virus RNAs and ALT levels in the sera. Experiments using a large number of gerbils may help overcome this weak point. Our future studies will focus on the pathogenicity of HEV in Mongolia gerbils.

## Figures and Tables

**Figure 1 viruses-14-01125-f001:**
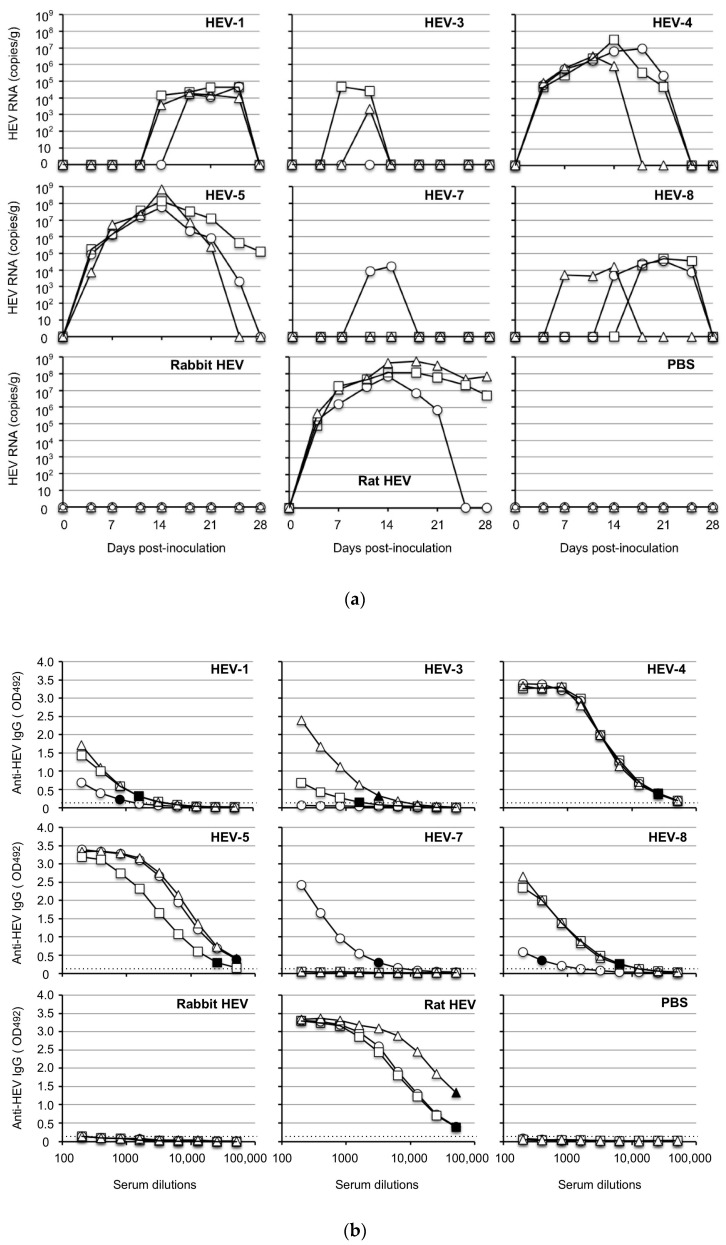
HEV replication in Mongolia gerbils. Twenty-seven Mongolia gerbils were randomly separated into nine groups (*n* = 3 per group). Individual gerbils are indicated by ○, △ and ☐ (**a**,**b**) or white, gray and black bars (**c**). Each group received HEV-1, HEV-3, HEV-4, HEV-5, HEV-7, HEV-8, rabbit HEV, rat HEV or phosphate-buffered saline (PBS) via intraperitoneal injection. The kinetics of the viral RNA in the fecal specimens were measured by RT-qPCR (**a**). The serum and tissue samples were collected at the end of the experiment (day 28 p.i.), and the anti-HEV-IgG antibody titers were determined using an ELISA with the VLPs of HEV-1 (to detect HEV-1, HEV-3, HEV-4, HEV-5, HEV-7, HEV-8 or rabbit HEV) or those of rat HEV (to detect rat HEV) as the antigens (**b**). Dotted lines: the cut-off values. The minimum endpoints of the antibody titers are blackened (●, ▲, ■) (**b**). The viral RNA titers in the liver, spleen and serum samples (**c**).

**Figure 2 viruses-14-01125-f002:**
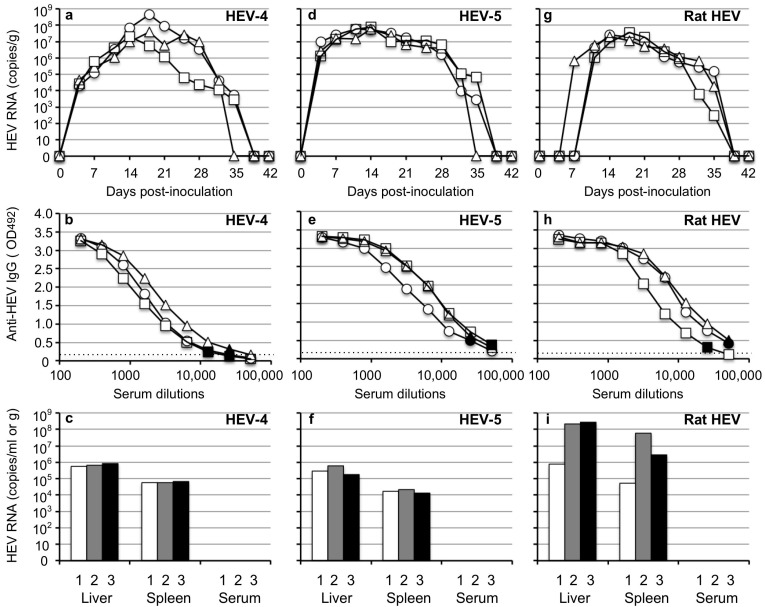
Infectivity of HEV discharged into the feces. Three groups of Mongolia gerbils (*n* = 3 per group) were inoculated with the 10% stool suspension prepared from the HEV-4-inoculated, HEV-5-inoculated or rat HEV-inoculated gerbils. Individual gerbils are indicated by ○, △ and ☐ (**a**,**b**,**d**,**e**,**g**,**h**) or white, gray and black bars (**c**,**f**,**i**). The kinetics of the viral RNA in the fecal specimens were determined by RT-qPCR (**a**,**d**,**g**). The serum and tissue samples were collected at the end of the experiment (day 42 p.i.). The serum anti-HEV IgG antibodies were measured using an ELISA with either VLPs of HEV-1 (to detect HEV-4 and HEV-5) or those of rat HEV (to detect rat HEV) as the antigens. Dotted lines: the cut-off values and the minimum endpoints of the antibody titers are blackened (●, ▲, ■) (**b**,**e**,**h**). The viral RNA titers detected in the liver, spleen and serum (**c,f,i**).

**Figure 3 viruses-14-01125-f003:**
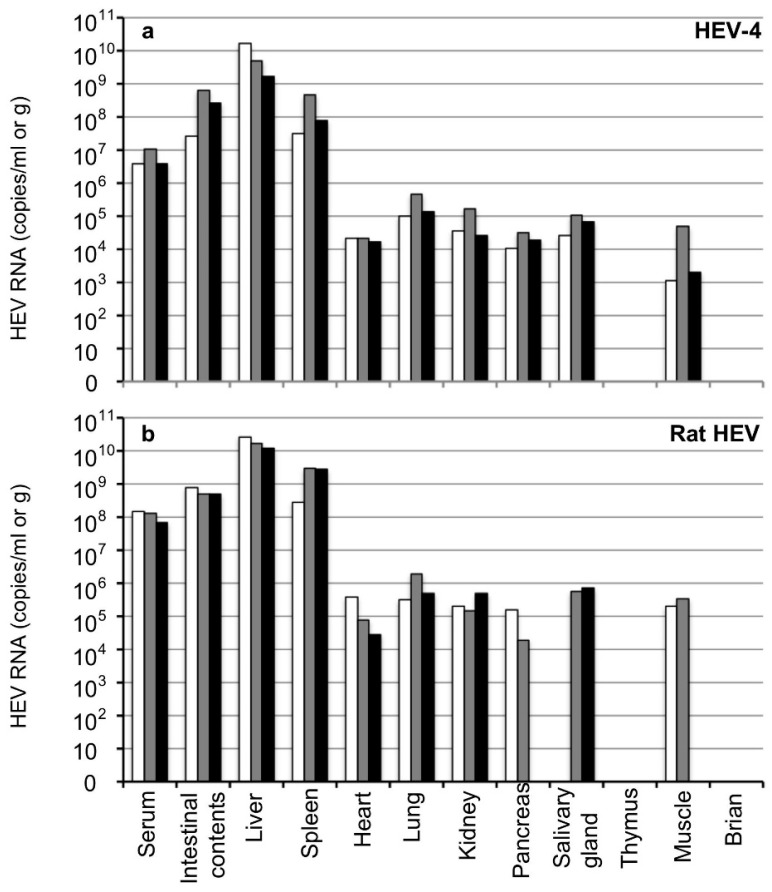
Detection of the viral RNA in the tissues in the early stage of infection. Six Mongolia gerbils were randomly separated into two groups (*n* = 3 per group). Individual gerbils are indicated by white, gray and black bars. One group was inoculated with HEV-4 (**a**) and the other with rat HEV (**b**) via intraperitoneal injection. All gerbils were euthanized on day 14 p.i. The serum samples, intestinal contents, liver, spleen, heart, lung, kidney, pancreas, salivary gland, thymus, muscle and brain were collected, and the viral RNA was measured by RT-qPCR.

**Figure 4 viruses-14-01125-f004:**
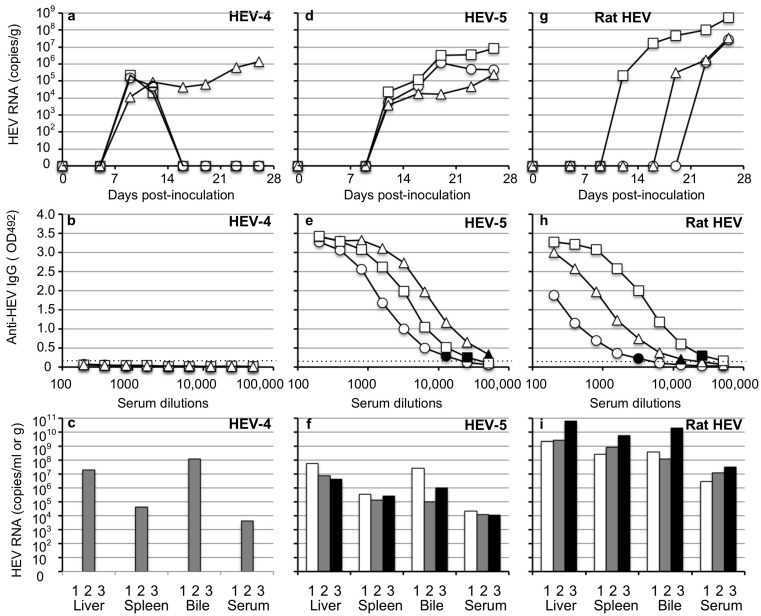
HEV-4, HEV-5 and rat HEV orally transmitted to Mongolia gerbils. Nine Mongolia gerbils were randomly separated into three groups (*n* = 3 per group). Individual gerbils are indicated by ○, △ and ☐ (**a**,**b**,**d**,**e**,**g**,**h**) or white, gray and black bars (**c**,**f**,**i**). Each group was orally inoculated with HEV-4, HEV-5 or rat HEV. The kinetics of the viral RNA in the fecal specimens were determined by RT-qPCR (**a**,**d**,**g**). The liver, spleen, bile and serum samples were collected at the end of the experiment (day 26 p.i.). The serum anti-HEV IgG antibodies were detected by using an ELISA with either the VLPs of HEV-1 (to detect HEV-4 and HEV-5) or those of rat HEV (to detect rat HEV) as the antigens. Dotted lines: the cut-off value and the minimum endpoints of the antibody titers were blackened (●, ▲, ■) (**b**,**e**,**h**). The viral RNA titers in the liver, spleen, bile and serum (**c**,**f**,**i**).

**Table 1 viruses-14-01125-t001:** ALT (IU/L) in the sera collected from Mongolia gerbils on day 28 p.i.

HEV Strain	Gerbil 1	Gerbil 2	Gerbil 3
HEV-1	45	66	47
HEV-3	54	49	69
HEV-4	63	69	78
HEV-5	59	58	78
HEV-7	46	46	63
HEV-8	54	66	60
Rabbit HEV	42	47	48
Rat HEV	57	62	59
PBS	53	57	60

**Table 2 viruses-14-01125-t002:** ALT (IU/L) in the sera collected from Mongolia gerbils on day 14 p.i.

HEV Strain	Gerbil 1	Gerbil 2	Gerbil 3
HEV-4	59	58	98
Rat HEV	92	70	61

**Table 3 viruses-14-01125-t003:** ALT (IU/L) in the sera collected from orally inoculated Mongolia gerbils on day 26 p.i.

HEV Strain	Gerbil 1	Gerbil 2	Gerbil 3
HEV-4	53	47	56
HEV-5	56	70	61
Rat HEV	56	50	59

## Data Availability

The sequences of HEV used in the study have been assigned (GenBank accession no. LC061267, AB740232, LC657084, AB573435, KJ496144, MH410176, LC484431 and JX120573).

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
