# Peer review of "Mongolia Gerbils Are Broadly Susceptible to Hepatitis E Virus"

_viruses, 2022, doi:10.3390/v14061125_

Round 1
Reviewer 1 Report
The authors did multiple work on the susceptibility of different genotypes of HEV using Mongolia Gerbils model. Here are some suggestions:
- Could the author explain why the serum level of ALT is not changed after HEV infection?
- Could the author show the viral genome sequencing data as supplementary materials?
- Are there any pathological changes in animals infected with HEV?
Author Response
Ms. ID viruses-1724016: "Mongolia gerbils are broadly susceptible to hepatitis E virus" by Zhang et al.
Responses to Reviewer 1:
The authors did multiple work on the susceptibility of different genotypes of HEV using Mongolia Gerbils model. Here are some suggestions:
- Could the author explain why the serum level of ALT is not changed after HEV infection?
Response:
As described in the Discussion, we collected the serum samples exclusively at the end of each experiment since the collection of blood from gerbils is very difficult, and we were unable to perform kinetic experiments to measure the virus RNAs and ALT levels in the sera. We do not have enough data to determine whether the level of ALT was changed during the infection. We hope that the experiments using a large number of gerbils may help us to understand the kinetic ALT level during HEV infection.
- Could the author show the viral genome sequencing data as supplementary materials?
Response:
The original nucleotide sequences of the HEV strains that we used in the present study have been submitted to the GenBank. Because no nucleotide sequence change was found in the viral genomes after infection, we do not think it is necessary to show the genome sequence as supplementary materials.
- Are there any pathological changes in animals infected with HEV?
Response:
The present study was focused on Mongolia gerbils' susceptibility to HEV, and we did not examine the pathological changes in liver or other tissues.

Reviewer 2 Report
In the manuscript, Zhang et al. have examined the competence of Mongolian gerbils for infecting HEV with the genotype of HEV-1, HEV-3, HEV-4, HEV-5, HEV-7, HEV-8, rabbit HEV, and rat HEV, confirming that gerbils are broadly susceptible to various HEV genotypes after monitoring the dissemination of virus in the liver, serum, intestinal contents, and other tissues. Authors suggested considerably distinct infectivity among the HEV genotypes and found that HEV-4, HEV-5 and rat HEV replicated well in Mongolia gerbil.
This manuscript is an interesting extension of a recently reported critical progression (PMID: 35230152), revealing that the Mongolian gerbil is an optimal small animal model for HEV infection and liver injury, including genotypes G1, G3 and G4. The current manuscript demonstrates again that the gerbil can serve as a valuable model in basic and translational HEV research, such as cross-protection and vaccine efficacy.
Overall experimental designs and execution are sound.
Minor comments:
- Introduction: the model of human liver chimeric mice should not be ignored.
- Line 101-103: Titers and administration amount of virus should be stated clearly. How the consistency among each gerbil is controlled?
- Line 160: Alanine aminotransferase (ALT) values in the rabbit sera were monitored weekly. Rabbit or gerbil?
- Table 2 and 3, the negative control is required.
- Line 347: Could the authors give a proper explanation, as obviously, viral RNA was detected in one gerbil?
Author Response
Ms. ID viruses-1724016: "Mongolia gerbils are broadly susceptible to hepatitis E virus" by Zhang et al.
Responses to Reviewer 2:
In the manuscript, Zhang et al. have examined the competence of Mongolian gerbils for infecting HEV with the genotype of HEV-1, HEV-3, HEV-4, HEV-5, HEV-7, HEV-8, rabbit HEV, and rat HEV, confirming that gerbils are broadly susceptible to various HEV genotypes after monitoring the dissemination of virus in the liver, serum, intestinal contents, and other tissues. Authors suggested considerably distinct infectivity among the HEV genotypes and found that HEV-4, HEV-5 and rat HEV replicated well in Mongolia gerbil.
This manuscript is an interesting extension of a recently reported critical progression (PMID: 35230152), revealing that the Mongolian gerbil is an optimal small animal model for HEV infection and liver injury, including genotypes G1, G3 and G4. The current manuscript demonstrates again that the gerbil can serve as a valuable model in basic and translational HEV research, such as cross-protection and vaccine efficacy.
Overall experimental designs and execution are sound.
Minor comments:
- Introduction: the model of human liver chimeric mice should not be ignored.
Response:
We have added the following text in accord with your suggestion:
"Although the human liver chimeric mice might be used as a small-animal model of HEV, the costs of using chimeric mice are high (ref)."
The related reference was also added.
- Line 101-103: Titers and administration amount of virus should be stated clearly. How
the consistency among each gerbil is controlled?
Response:
We revised the sentences as follows:
"Five milliliters (107 copies/mL) of the cell culture supernatant of HEV-4 and HEV-5 or 10% stool suspension of rat HEV was mixed with 15 mL of drinking water and then orally administered to each gerbil daily for 5 consecutive days. The administration amount of the virus was determined daily, and we confirmed that the virus mixtures were completely consumed by the gerbils."
- Line 160: Alanine aminotransferase (ALT) values in the rabbit sera were monitored weekly. Rabbit or gerbil?
Response:
We have corrected the sentence as follows:
"Alanine aminotransferase (ALT) values in the gerbil sera were monitored using a Fuji Dri-Chem Slide GPT/ALT-PIII kit (Fujifilm, Saitama, Japan)."
- Table 2 and 3, the negative control is required.
Response:
Since the ALT level in three Mongolia gerbils used as negative controls were described in Table 1, we did not present the negative controls in Table 2 and Table 3.
- Line 347: Could the authors give a proper explanation, as obviously, viral RNA was detected in one gerbil?
Response:
When the gerbils were inoculated with HEV-7, only one animal was positive for HEV-7 RNA; we speculate that Mongolia gerbils have low susceptibility to HEV-7 and showed different sensitivity among the individual gerbils.
